# Comparison of Clonogenic Survival Data Obtained by Pre- and Post-Irradiation Methods

**DOI:** 10.3390/jpm10040171

**Published:** 2020-10-15

**Authors:** Takahiro Oike, Yuka Hirota, Narisa Dewi Maulany Darwis, Atsushi Shibata, Tatsuya Ohno

**Affiliations:** 1Department of Radiation Oncology, Graduate School of Medicine, Gunma University, 3-39-22, Showa-machi, Maebashi 371-8511, Japan; yukahirota@gunma-u.ac.jp (Y.H.); m1920021@gunma-u.ac.jp (N.D.M.D.); tohno@gunma-u.ac.jp (T.O.); 2Heavy Ion Medical Center, Gunma University, 3-39-22, Showa-machi, Maebashi 371-8511, Japan; 3Department of Radiation Oncology, Faculty of Medicine Universitas Indonesia—Dr. Cipto Mangunkusumo Hospital, Jl. P. Diponegoro no. 71, Jakarta 10430, Indonesia; 4Initiative for Advanced Research (GIAR), Gunma University, 3-39-22, Showa-machi, Maebashi 371-8511, Japan; shibata.at@gunma-u.ac.jp

**Keywords:** clonogenic assays, methods, plating, cancer, radiation, radiosensitivity

## Abstract

Clonogenic assays are the gold standard to measure in vitro radiosensitivity, which use two cell plating methods, before or after irradiation (IR). However, the effect of the plating method on the experimental outcome remains unelucidated. By using common cancer cell lines, here we demonstrate that pre-IR and post-IR plating methods have a negligible effect on the clonogenic assay-derived photon sensitivity as assessed by SF_2_, SF_4_, SF_6_, SF_8_, D_10_, or D_50_ (N.B. SFx indicates the survival at X Gy; Dx indicates the dose providing X% survival). These data provide important biological insight that supports inter-study comparison and integrated analysis of published clonogenic assay data regardless of the plating method used.

## 1. Introduction

Clonogenic assays are the gold standard method for assessing radiosensitivity in vitro [1]. Cancer research studies have reported the radiosensitivity of various cell lines obtained using clonogenic assays [2]. In addition, clonogenic assays are used to determine the relative biological effectiveness (RBE) of carbon ions over photons in clinical carbon ion radiotherapy (CIRT) [3,4,5].

Recent advances in computer science have enabled the integration of published experimental data into big data platforms. For example, in the field of genomics, published sequencing data are compiled in databases such as the Catalogue of Somatic Mutations in Cancer [6] and Cancer Cell Line Encyclopedia (CCLE) [7]. From this perspective, integration of published radiosensitivity data obtained by clonogenic assays will be a powerful strategy to promote radiation oncology research [8,9,10]. However, the data integration is difficult because there are two types of clonogenic assays using different cell plating methods; namely, cells are plated before or after irradiation (referred to hereafter as pre-IR plating and post-IR plating, respectively). In pre-IR plating methods, single cells in suspensions are seeded on plates and subjected to a treatment of interest (e.g., drug exposure or irradiation) after additional incubation for a few hours to days that allows cells to attach on the plates. Pre-IR methods are capable of creating multiple technical replicates for a treatment of interest easily; therefore, pre-IR plating methods are predominantly used for cancer research [9]. On the other hand, in post-IR plating methods, plates with subconfluent cells are subjected to a treatment of interest, which is followed by trypsinization, single cell suspension, and seeding for replication. Post-IR plating methods have been utilized in the historical work on the beam design of CIRT [3,4,5]. This is probably due to the fact that carbon ions are an extremely limited medical resource as there are only 13 institutions available for CIRT across the world as reported on the website of Particle Therapy Co-Operative Group (https://www.ptcog.ch/index.php/facilities-in-operation). We assume that researchers have intended to save precious machine time by avoiding irradiation of all replicates.

In principle, pre-IR and post-IR methods are different in terms of cell condition at the time of irradiation. In post-IR methods, cells are capable of cell-to-cell signal transduction in immediate response to irradiation before being separated from each other, which may affect radiosensitivity. However, the effect of the plating method on the experimental outcomes of clonogenic assays remains unelucidated, which limits progress in radiation oncology research. For example, Amornwichet et al. reported that the RBE of carbon ions in nine *epidermal growth factor receptor (EGFR)-*wild-type non-small cell lung carcinoma (NSCLC) cell lines was 2.6 ± 0.3 at the center of the 6-cm-wide spread-out Bragg peak (SOBP) [11], whereas Kagawa et al. reported that the RBE in human salivary gland (HSG) cells, the cell line used as the reference in the clinical CIRT beam set-up, was 1.8 at the center of the same 6-cm SOBP [5]. Although these data indicate that CIRT is more effective for *EGFR* wild-type NSCLCs, a definitive conclusion cannot be made because the former and latter studies used pre-IR and post-IR plating methods, respectively.

The aim of this study was to elucidate the effect of the clonogenic assay plating method on the experimental outcome of cancer cell radiosensitivity.

## 2. Materials and Methods

### 2.1. Cell Line and Cell Culture

A549 (human lung adenocarcinoma cell line) and HSG (human salivary gland tumor cell line) were used in this study. A549 was chosen because this cell line is commonly used for clonogenic assays in general cancer research, which predominantly uses pre-IR plating methods [2,9]. HSG was chosen because this cell line has been used as the reference cell line for CIRT beam set-up, which uses post-IR plating methods [3,4,5]. Previous studies indicate that both cell lines show intermediate-to-relatively-low sensitivity to photons [11,12,13]. A549 cells were purchased from ATCC (CCL-185, Manassas, VA, USA). HSG cells were purchased from JCRB Cell Bank (HSGc-C5, JRCB1070, Ibaragi, Japan). Cells were cultured in RPMI-1640 (Sigma-Aldrich, St. Louis, MO, USA) supplemented with 10% fetal bovine serum (Life Technologies, Carlsbad, CA, USA) in a 5% CO_2_ incubator at 37 °C. No other additives were used in the media. Cells in the log-phase of growth were used for experiments.

### 2.2. Clonogenic Assays

Clonogenic assays were performed as described previously [1]. For a given assay, either pre-IR or post-IR plating methods were employed.

For pre-IR plating, cells were detached from culture dishes using trypsin (Sigma-Aldrich) and prepared as single cell suspensions in culture medium. The cells were counted using a hemocytometer under an inverted microscope. The single cell suspensions were subjected to two serial dilutions at 1:10 (i.e., 1:100 dilution in total), and the resulting suspensions were used for plating. The plated cells were incubated for a minimum period to enable cell attachment (approximately 6 h), and were exposed to X-ray irradiation at 2, 4, 6, and 8 Gy, or were sham-irradiated.

For post-IR plating, cells were trypsinized, and 2 × 10^5^ cells were plated on a 3.5-cm dish. After incubation for 48 h, and when the cells reached 80–90% confluency, the cells were exposed to X-ray irradiation at 2, 4, 6, and 8 Gy, or were sham-irradiated. Immediately after irradiation, the cells were trypsinized and prepared as single cell suspensions in culture medium. The cells were counted using a hemocytometer under an inverted microscope. The single cell suspensions were subjected to two serial dilutions at 1:10, and the resulting suspensions were used for plating.

For all experiments, the cells were incubated for an additional 12 days, fixed with methanol, and stained with crystal violet. Colonies comprising ≥50 cells were counted under an inverted microscope. The surviving fraction at a given dose point was calculated by dividing the number of colonies by the number of seeded cells, which was further divided by plating efficiency calculated based on unirradiated controls. The surviving fraction at X Gy is referred to hereafter as SF_X_. SF_2_, SF_4_, SF_6_, and SF_8_ were fitted to the linear quadratic model [12], from which D_10_ and D_50_ (i.e., the doses decreasing cell survival to 10% and 50%, respectively) were calculated. For both pre-IR and post-IR plating methods and for both cell lines, the number of cells plated per well was unified as 200, 200, 200, 400, and 400 for 0, 2, 4, 6, and 8 Gy, respectively. Experiments were repeated three times. Four samples were used for each experiment.

### 2.3. Irradiation

X-ray irradiation was performed using an MX-160Labo (160 kVp, 1.06 Gy/min; mediXtec, Matsudo, Japan) [4].

### 2.4. Statistics

Differences between two groups were examined using the non-parametric two-sided Mann–Whitney U-test. Differences were considered statistically significant at *p* < 0.05. All statistical analyses were performed using Prism8 (GraphPad Software, San Diego, CA, USA).

## 3. Results

To evaluate the effect of different plating methods on the experimental outcomes of clonogenic assays, we performed clonogenic assays using pre-IR or post-IR plating methods, while keeping the other experimental settings constant. The radiosensitivity endpoints commonly used in this field, i.e., SF_2_, SF_4_, SF_6_, SF_8_, D_10_, and D_50_, were compared between the two methods [9]. Plating efficiency exceeded 60% in all experiments, with a median of 82%. In the assessment of SF_2_, SF_4_, SF_6_, D_10_, and D_50_, the coefficient of variation (CV) among three independent experiments was <20% in all experimental settings (median, 7%; 1–17%); these values were sufficiently low compared with previously published data [2,9]. The CV values for SF_8_ were relatively high (median, 27%; 6–47%); nevertheless, these values were still lower than those published by Nuryadi et al., who calculated the CV for SF_8_ in A549 cells from 20 repeated experiments using the same protocol in the same laboratory [2]. These data suggest that the experiments in this study were performed in a technically sound manner.

In A549 cells, no statistically significant differences in the outcomes were observed between pre-IR and post-IR plating methods for SF_2_, SF_4_, SF_6_, SF_8_, D_10_, and D_50_ (Figure 1a–b). Survival plots demonstrated a high consistency between the two plating methods (Figure 1c–e).

In HSG cells, no statistically significant differences in the outcomes were observed between pre-IR and post-IR plating methods for SF_2_, SF_4_, SF_6_, SF_8_, D_10_, and D_50_ (Figure 2a–b). Survival plots demonstrated a high consistency between the two plating methods (Figure 2c–e). Taken together, these data suggest that the influence of the difference in the plating methods on the outcomes of clonogenic assays is negligible in A549 and HSG cells.

## 4. Discussion

To the best of our knowledge, this is the first study to investigate the effect of different plating methods of clonogenic assays on the experimental outcomes. The results provide important insight supporting inter-study comparisons and integrated analysis of published clonogenic assay data regardless of the plating method used, which will contribute the promotion of radiation oncology research in the era of big data science.

The concept of precision medicine, that is optimization of treatment strategy based on genetic information of individual cancers, has become widespread in the field of cancer chemotherapy according to the advancement of next-generation sequencers. For example, if a lung cancer was found to harbor *ret proto-oncogene (RET)* fusions, then the cancer can be efficiently treated with Vandetanib, an inhibitor of RET tyrosine kinase [14]. In the field of radiation oncology on the other hand, the concept of precision medicine has not been applied to the clinic sufficiently. Theoretically, if we could predict the sensitivity of a tumor to radiotherapy at the time of diagnosis, then we can stratify radioresistant cases to the radiotherapy modalities capable of high-dose delivery (e.g., stereotactic body radiotherapy and particle therapies) that are rarer than conventional three-dimensional conformal radiotherapy. To this end, establishment of genetic profiles that predict cancer radioresistance is an urgent need. One of the barriers for the research aiming to meet this need is the absence of the big data pertaining to cancer cell radiosensitivity that can be used for analysis in combination with genomics data. This is in contrast to the situation for chemotherapy, where multiple databases for the sensitivity of cancer cells to anticancer drugs (e.g., CCLE) are open to public [7]. Although we find an enormous number of publications that report on the cancer cell radiosensitivity as assessed by clonogenic assays, the variance in the plating method has prevented us from conducting inter-study comparison and integration of these radiosensitivity data. Our data provide insight in overcoming this issue; using multiple cancer cell lines commonly used in this field, we showed that the difference in the plating method on the clonogenic assay-derived radiosensitivity data is negligibly small in A549 and HSG cells, suggesting that the published clonogenic data can be analyzed in combination regardless of the plating method. In addition, as explained in the Introduction, our data rationalize the inter-translation between genomics-associated radiosensitivity data and carbon-ion RBE data obtained predominantly using pre-IR and post-IR plating methods, respectively. Additionally, we assume that the findings from this study may be applicable to carbon ion experiments, warranting further research.

This study had several limitations. Minor subtypes of plating methods, such as IR in cell suspensions or delayed post-IR plating [9], were not investigated. In addition, cell lines other than A549 and HSG were not included. Research is warranted to further elucidate the influence of the difference in the methods in clonogenic assays on cancer cell radiosensitivity.

## 5. Conclusions

We showed that SF_2_, SF_4_, SF_6_, SF_8_, D_10_, and D_50_ values obtained using clonogenic assays were highly consistent between pre-IR and post-IR methods in A549 and HSG cells. These data support the strategic robustness of inter-study comparisons and integrated analysis of published clonogenic assay data, regardless of the plating method used. Thus, these data will contribute to promote radiation oncology research in the era of big data science.

## Figures and Tables

**Figure 1 jpm-10-00171-f001:**
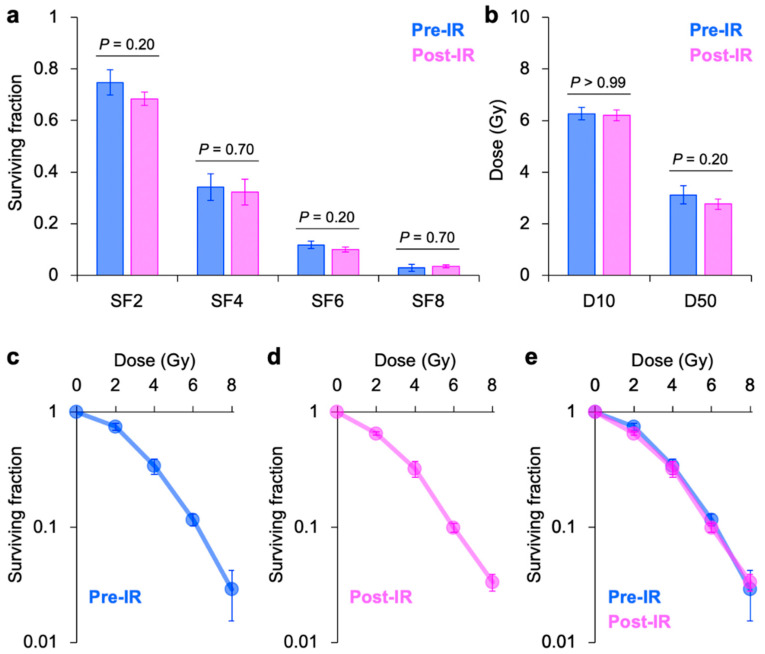
Clonogenic survival of X-ray-irradiated A549 cells assessed using plating methods in which cells are plated before (pre-IR) or after (post-IR) irradiation. (**a**) SF_2_, SF_4_, SF_6_, and SF_8_. (**b**) D_10_ and D_50_. *p*-values were determined using the Mann–Whitney U-test. (**c**) Survival plots from pre-IR-plated cells. (**d**) Survival plots from post-IR-plated cells. (**e**) Survival plots from pre-IR (blue) or post-IR (violet) plating methods. Graphs are presented in 50% translucent colors; therefore, purple color indicates an overlap between the two plating methods. SFx indicates the survival at X Gy; Dx indicates the dose providing X% survival. Error bars indicate standard deviation.

**Figure 2 jpm-10-00171-f002:**
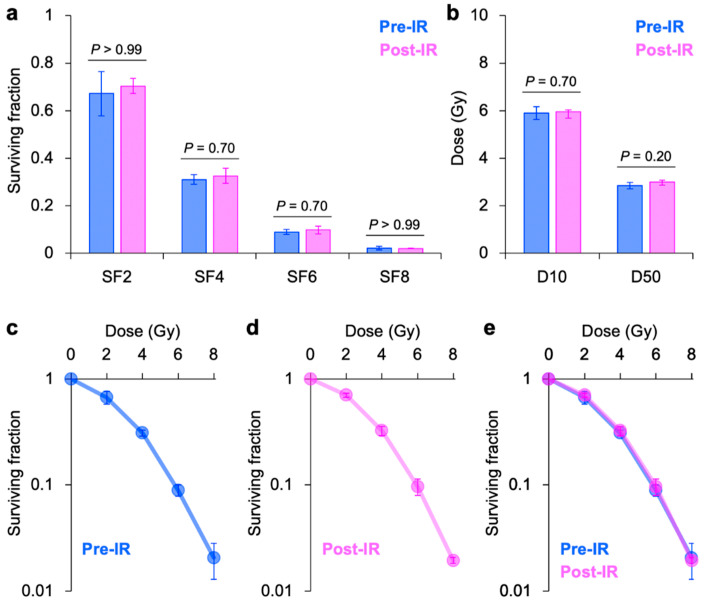
Clonogenic survival of X-ray-irradiated human salivary gland (HSG) cancer cells assessed using pre-IR or post-IR plating methods. (**a**) SF_2_, SF_4_, SF_6_, and SF_8_. (**b**) D_10_ and D_50_. *p*-values were determined using the Mann–Whitney U-test. (**c**) Survival plots from pre-IR-plated cells. (**d**) Survival plots from post-IR-plated cells. (**e**) Survival plots from pre-IR (blue) or post-IR (violet) plating methods. Graphs are presented in 50% translucent colors; therefore, purple color indicates an overlap between the two plating methods. SFx indicates the survival at X Gy; Dx indicates the dose providing X% survival. Error bars indicate standard deviation.

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
