# Peer review of "Comparison of Clonogenic Survival Data Obtained by Pre- and Post-Irradiation Methods"

_jpm, 2020, doi:10.3390/jpm10040171_

Round 1
Reviewer 1 Report
The paper contains new results which are useful for comparison and analysis of experimental data obtained by different methods (namely, by the so-called pre-IR plating and post-IR plating methods) and, eventually, can be useful for optimization of several scenarios of radiation cancer therapy. Publication of the paper will be possible after minor revision:
- The authors should describe briefly the effects which could cause, in principle, the difference of the results obtained when using pre-IR plating and post-IR plating methods.
- The authors should either explain, at least briefly, their statement about applicability of their results for analysis of the data obtained in the experiments with carbon ions or present this statement as the assumption.
- Several sentences should be edited. (3a) I do not understand the use of the word "are" in the sentence on lines 43-44 (page 1). (3b) The source(s) of the important statement " there are only 13 institutions available for CIRT over the world" (line 45, page 2) should be mentioned. (3c) The second part of the sentence containing this statement ("it is assumed that researchers have intended to save precious machine time by avoiding irradiation of all replicates" should be edited. If such assumption was done earlier, the reference(s) should be presented. If this assumption is the new one made by the authors, they should write "we assume that...", or "it is possible that", or something similar. (3d) All of the abbreviations, including "EGFR" and " HSG" should be explained (I would like to note that the authors explain the well-known abbreviation "SOBP", and this, from my point of view, is correct). (3e) The words "Although these data indicate that CIRT is effective for EGFR wild-type NSCLCs..." (lines 52-53, page 2) are not quite clear, I assume that the word "more" should be inserted between "is" and "effective".
Author Response
Reviewer 1
The paper contains new results which are useful for comparison and analysis of experimental data obtained by different methods (namely, by the so-called pre-IR plating and post-IR plating methods) and, eventually, can be useful for optimization of several scenarios of radiation cancer therapy. Publication of the paper will be possible after minor revision.
Response:
We sincerely thank the reviewer for evaluating our manuscript and for the encouraging comments. According to the suggestion, we thoroughly revised the manuscript as follows.
The authors should describe briefly the effects which could cause, in principle, the difference of the results obtained when using pre-IR plating and post-IR plating methods.
Response:
We sincerely thank the reviewer for the critical comment. In principle, pre-IR and post-IR methods are different in terms of cell condition at the time of irradiation. In post-IR methods, cells are capable of performing cell-to-cell signal transduction in immediate response to irradiation before being separated with each other, which may affect radiosensitivity. This was added in lines 49–52.
The authors should either explain, at least briefly, their statement about applicability of their results for analysis of the data obtained in the experiments with carbon ions or present this statement as the assumption.
Response:
We sincerely thank the reviewer for the insightful comment. According to the suggestion, the following sentence was added: "we would assume that the findings from this study may be applicable to carbon ion experiment, warranting further research" (lines 172–174).
Several sentences should be edited.
(3a) I do not understand the use of the word "are" in the sentence on lines 43-44 (page 1).
Response:
We apologize. This was a typo. "Are" was deleted (line 44).
(3b) The source(s) of the important statement "there are only 13 institutions available for CIRT over the world" (line 45, page 2) should be mentioned.
Response:
We thank the reviewer for the comment. The source of the statement, i.e., website of the Particle Therapy Co-Operative Group, was added (line 46–47).
(3c) The second part of the sentence containing this statement ("it is assumed that researchers have intended to save precious machine time by avoiding irradiation of all replicates" should be edited. If such assumption was done earlier, the reference(s) should be presented. If this assumption is the new one made by the authors, they should write "we assume that...", or "it is possible that", or something similar.
Response:
We thank the reviewer for the advice. The sentence was changed to "We assume that..." (line 47–48).
(3d) All of the abbreviations, including "EGFR" and " HSG" should be explained (I would like to note that the authors explain the well-known abbreviation "SOBP", and this, from my point of view, is correct).
Response:
We thank the reviewer for the comment. EGFR was spelled out as "epidermal growth factor receptor" (line 54–55). HSG was spelled out as "human salivary gland" (line 65). In addition, RET was spelled out as "ret proto-oncogene " (line 153).
(3e) The words "Although these data indicate that CIRT is effective for EGFR wild-type NSCLCs..." (lines 52-53, page 2) are not quite clear, I assume that the word "more" should be inserted between "is" and "effective".
Response:
We thank the reviewer for the suggestion. Accordingly, "more" was inserted between "is" and "effective" (line 58).

Reviewer 2 Report
Oike et al compare the clonogenic survival following plating pre- or post-irradiation of two different cell lines. The authors note no significant difference between plating methods for these cell lines following treatment with irradiation.
- Please define SF2, SF4, SF6, SF8, D10, and D50 within the abstract and the figure legends.
- pg 1, Line 41: Remove “one”
- Pg 1, Line 43: “works” should be “work”
- Pg 1, line 44: Remove the first “are”
- Pg 2, line 45: “over” should be changed to “across”
- Please confirm that no other additives were used in the cell media (eg NEAA or Pen/Step).
- Please describe what these cell lines are (eg where the originated) and why they were selected for this study. Are the lines radiosensitive or resistant?
- Please provide the number of cells for each cell line plated in the different treatment groups. Were the same number of cells plated for each plating method?
- Pg 3, line 112 and Pg 4, line 122: replace “showed” with “revealed” or “demonstrated” or something similar
- Please state what the error bars indicate (eg standard deviation vs error).
- In the last line of the results section and in the discussion, broad statements regarding the negligible influence on plating methods are used. It may be more appropriate to add that these results were observed for these specific cell lines.
- Pg 5, line 144: Should the sentence read “has not been applied to the clinic well”?
- Pg 5, line 154: Should the sentence read “has prevented us from conducting inter-study”?
- Pg 5, line 155: Should the sentence read “Our data provide insight in overcoming this issue”?
- Pg 5, line 159: add “the” before “Introduction”
Author Response
Reviewer 2
Oike et al. compare the clonogenic survival following plating pre- or post-irradiation of two different cell lines. The authors note no significant difference between plating methods for these cell lines following treatment with irradiation.
Response:
We sincerely thank the reviewer for evaluating our manuscript. According to the suggestion, we thoroughly revised the manuscript as follows.
Please define SF2, SF4, SF6, SF8, D10, and D50 within the abstract and the figure legends.
Response:
We sincerely thank the reviewer for the suggestion. Accordingly, the following sentence was added in lines 19–20, 128–129, and 142: "SFx indicates the survival at X Gy; Dx indicates the dose providing X% survival".
Pg 1, Line 41: Remove “one”.
Response:
We apologize for the typo. "One" was removed (line 42).
Pg 1, Line 43: “works” should be “work”.
Response:
We thank the reviewer for the comment. Accordingly, "works" was changed to "work" (line 44).
Pg 1, line 44: Remove the first “are”.
Response:
We apologize for the typo. "Are" was deleted (line 44).
Pg 2, line 45: “over” should be changed to “across”.
Response:
We thank the reviewer for the comment. Accordingly, "over" was changed to "across" (line 46).
Please confirm that no other additives were used in the cell media (eg NEAA or Pen/Step).
Response:
We conform that no other additives were used in the cell media. This was added in lines 73–74.
Please describe what these cell lines are (e.g., where the originated) and why they were selected for this study. Are the lines radiosensitive or resistant?
Response:
We thank the reviewer for the important comment. The information was added in lines 65–70 as follows: "A549 (human lung adenocarcinoma cell line) and HSG (human salivary gland tumor cell line) were used in this study. A549 was chosen because this cell line is commonly used for clonogenic assays in general cancer research, which predominantly uses pre-IR plating methods [references #2, #9]. HSG was chosen because this cell line is used as the reference cell line for CIRT beam set-up, which uses post-IR plating methods [references #3–5]. Previous studies indicate that both cell line show intermediate to relatively-low sensitivity to photons [references #11–13].
Please provide the number of cells for each cell line plated in the different treatment groups. Were the same number of cells plated for each plating method?
Response:
We sincerely thank the reviewer for the critical comment. For both pre-IR and post-IR plating methods and for both cell lines, the number of cells plated per well was unified as 200, 200, 200, 400, and 400 for 0, 2, 4, 6, and 8 Gy, respectively. This was added in lines 96–98.
Pg 3, line 112 and Pg 4, line 122: replace “showed” with “revealed” or “demonstrated” or something similar.
Response:
We thank the reviewer for the comment. According to the suggestion, "show" was replaced to "demonstrated" (lines 122 and 133).
Please state what the error bars indicate (e.g., standard deviation vs error).
Response:
We thank the reviewer for the comment. Error bars indicate standard deviation. This was clarified in lines 128–129 and 142–143.
In the last line of the results section and in the discussion, broad statements regarding the negligible influence on plating methods are used. It may be more appropriate to add that these results were observed for these specific cell lines.
Response:
We thank the reviewer for the comment. According to the suggestion, the phrase "in A549 and HSG cells" was added in the last line of the results section and in the discussion (lines 135 and 168–169).
Pg 5, line 144: Should the sentence read “has not been applied to the clinic well”?
Response:
We apologize for the grammatical mistake. According to the comment, the phrase was corrected to "has not been applied to the clinic well" (line 155).
Pg 5, line 154: Should the sentence read “has prevented us from conducting inter-study”?
Response:
We apologize for the grammatical mistake. According to the comment, the phrase was corrected to "has prevented us from conducting inter-study" (line 165).
Pg 5, line 155: Should the sentence read “Our data provide insight in overcoming this issue”?
Response:
We apologize for the grammatical mistake. According to the comment, the phrase was corrected to "our data provide insight in overcoming this issue" (lines 166–167).
Pg 5, line 159: add “the” before “Introduction”
Response:
We apologize for the grammatical mistake. According to the comment, "the" was added before "Introduction" (line 170).

Reviewer 3 Report
Oike et al. report that their research supports inter-study comparison and integrated analysis of published clonogenic assay data regardless of the plating method used. The results are important and interesting but require further research.
Major
The authors presented the results of preliminary study performed on only two cell lines and should compare them with the results on other cell lines commonly used for clonogenic assays.
The authors themselves emphasize the shortcomings of their research, including the lack of investigation of IR in cell suspensions or delayed post-IR plating.
Author Response
Reviewer 3
Oike et al. report that their research supports inter-study comparison and integrated analysis of published clonogenic assay data regardless of the plating method used. The results are important and interesting but require further research. The authors presented the results of preliminary study performed on only two cell lines and should compare them with the results on other cell lines commonly used for clonogenic assays. The authors themselves emphasize the shortcomings of their research, including the lack of investigation of IR in cell suspensions or delayed post-IR plating.
Response:
We sincerely thank the reviewer for evaluating our manuscript. According to the criticism, the description of the last sentence in the Results was mitigated as follows: "these data suggest that the influence of the difference in the plating methods on the outcomes of clonogenic assays is negligible in A549 and HSG cells". Since this is Communication article, we believe that our data using two cell lines are acceptable for publication with the current revision.

Round 2
Reviewer 3 Report
The authors presented the results of preliminary study performed on only two cell lines and should compare them with the results on other cell lines commonly used for clonogenic assays.